# Metastable hybridization-based DNA information storage to allow rapid and permanent erasure

Jangwon Kim[1], Jin H. Bae [1], Michael Baym [2,3] & David Yu Zhang[1,4✉]

The potential of DNA as an information storage medium is rapidly growing due to advances in DNA synthesis and sequencing. However, the chemical stability of DNA challenges the complete erasure of information encoded in DNA sequences. Here, we encode information in a DNA information solution, a mixture of true message- and false message-encoded oligo-nucleotides, and enables rapid and permanent erasure of information. True messages are differentiated by their hybridization to a "truth marker" oligonucleotide, and only true messages can be read; binding of the truth marker can be effectively randomized even with a brief exposure to the elevated temperature. We show 8 separate bitmap images can be stably encoded and read after storage at 25 °C for 65 days with an average of over 99% correct information recall, which extrapolates to a half-life of over 15 years at 25 °C. Heating to 95 °C for 5 minutes, however, permanently erases the message.

[1] Department of Bioengineering, Rice University, Houston, TX 77030, USA. [2] Department of Biomedical Informatics, Harvard Medical School, Boston, MA 02115, USA. [3] Center for Theoretical Biological Physics, Rice University, Houston, TX 77005, USA. [4] Systems, Synthetic, and Physical Biology, Rice University, Houston, TX 77005, USA. ✉email: dyz1@rice.edu

Reliable and high-density storage of information is an emerging problem due to the rapid expansion of the scale of big data. DNA has been proposed as a promising alternative medium for durable and dense information storage[1–4], with demonstrations of stored information ranging up to 200 Mb[5,6]. Currently, the cost of writing (synthesizing) and reading (sequencing) information in DNA exceeds that of conventional methods such as flash drives and tape by many orders of magnitude[7], but biomedical research in genomics and synthetic biology are resulting in exponentially decreasing prices of DNA synthesis and sequencing.

The long half-life of DNA-based information storage has been frequently cited as an advantageous feature, when applied to important information such as court records. However, the high durability of DNA presents a significant challenge in the potential erasure of information that is both highly important and highly confidential, such as personal genomic information or military weapons designs. Conventional methods to destroy DNA include irradiating with ultraviolet light, using enzymes such as DNAse I[8], applying high temperature of over 200 °C[9,10], or using bleach[11]. These DNA destruction methods vary in their approach, but are generally difficult to implement in decentralized settings without specialized equipment, and erasure may not always complete within a reasonable timeframe (Supplementary Note 6)[10,11]. For example, to prevent interception during physical transport of a DNA information solution from one location to another, it may be necessary to have a rapid and complete method for erasing information without specialized equipment for fragile enzymes.

Here, we present a method of encoding information in DNA as a metastable aqueous solution that allows for rapid and permanent information erasure through a simple heating process (Fig. 1a). Compared to other erasure methods, it is much easier and faster to implement permanent information erasure in decentralized settings. In our approach, every file address includes a true message and at least one false message, and the messages are distinguished via the hybridization to a truth marker oligonucleotide (Fig. 1b). The stability of DNA hybridization is highly temperature sensitive, with extremely rapid dissociation of a DNA duplex at temperatures above its melting temperature[12,13] (Fig. 1c). Once the initial truth information is lost through heat-based dissociation, the original information cannot be recovered.

In contrast to conventional approaches where DNA information is stored only in its primary sequence, our approach combines potential messages encoded in DNA sequence with a true/false flag encoded in the metastable secondary structure of multi-stranded DNA hybridization. Erasure of information, in our context, does not require destroying the potential messages by chemical degradation of DNA, but rather relies on the randomization of true/false flags encoded in hybridization state. Given a DNA pool with $N$ addresses and $M$ potential messages at each address, there are $M^N$ possible files; the hybridization states of the $N \cdot M$ DNA oligos inform which one of the $M^N$ messages is the intended message. Once hybridization state information is lost, it is impossible to identify the original message among all the possibilities. For $M = 8$ and $N = 100$, the number of possible encoded messages is $2 \times 10^{90}$, far more than the number of atoms in the universe (estimated to be $10^{80}$).

In this paper, we experimentally demonstrate the hybridization-based DNA encoding of 8 bitmap images of famous artwork, and show that images can be stably encoded for 2 months at room temperature with no significant loss of information compared to a freshly prepared solution. A brief 5-min exposure to 95 °C, however, permanently erases the images and results in noise that is indistinguishable across the eight erased images. Thus, hybridization-based DNA information encoding simultaneously allows long-term storage of important

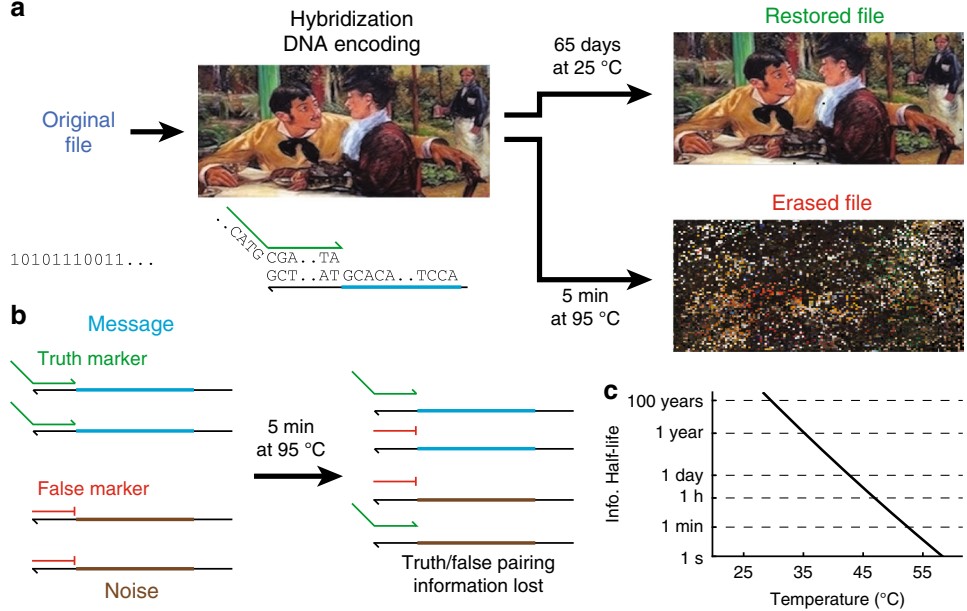

**Fig. 1 Overview of approach.** The artworks used are low-resolution images freely available for reuse under the terms of a Creative Commons license or that are marked as being in the Public Domain. **a** A computer file is encoded as a DNA solution, which can be stored stably for extended periods of time at room temperature or lower, but is quickly and permanently erased upon exposure to elevated temperatures (e.g., 95 °C). **b** Implementation of hybridization-based truth encoding. True messages intended to be part of the communicated information (blue) are prehybridized to truth markers, and false messages are prehybridized to false markers. Upon heating, the hybridization of the truth markers to the intended messages is disrupted. Subsequent cooling to room temperature would result in a random association of truth markers to messages and noise, and the information regarding which molecules correspond to true messages vs. noise is permanently lost. **c** The half-life of the truth marker hybridization is strongly temperature dependent. Plotted here is the expected half-life of a 20 nt DNA binding at different temperatures, based on the two-state model of DNA binding and published DNA thermodynamics parameters.

information and rapid erasure of sensitive information. Here we use the bitmap artwork images to demonstrate our approach in a way that is easy to visualize; however, we do not require any long-range pattern in the underlying data, and we imagine that practical uses of this approach would be on compressed information files with high K-complexity.

## Results

**Overall scheme and design of experiments**. In our method for DNA information encoding, true messages and false messages both possess a common marker binding region B (Fig. 2a). Two separate DNA oligonucleotides, the truth marker and the false marker, both have sequences complementary to region B; thus the truth marker and the false marker are able to bind to both the true message and the false message. The false marker, unlike the truth marker, has a 3′ chemical functionalization that prevents extension by DNA polymerase. The intended (true) messages are prehybridized with the truth marker, and the noise (false) messages are prehybridized with the false marker; these two solutions are combined to form the information solution. The information solution is read through enzymatic extension and amplification of the truth markers and subsequent sequencing-by-synthesis (NGS), as shown in Fig. 2b. Importantly, any DNA molecule with sequence satisfying our expected format can be used either as a true message or a false message, and furthermore may be used as both a true message for one information solution and as a false message for a different information solution.

The information solution is metastable and in a state of lowered entropy, because at equilibrium the truth markers and false markers will hybridize ratiometrically with the true and false messages. There are two possible reaction pathways for the rearrangement of truth marker binding partners: (1) a first-order pathway in which the truth marker spontaneously dissociates and then binds to a free message molecule, and (2) a second-order pathway in which a truth marker/true message complex reacts with a false marker/false message complex and exchange partners via strand displacement (see also Supplementary Note 1). When the information solution is heated to above the melting

temperature of the truth marker (e.g., 95 °C), the activation energy for mechanism (1) becomes very low, and the system quickly relaxes to equilibrium.

At lower temperatures, the activation energies for both mechanisms (1) and (2) are high, so the pairing information between truth markers and true messages can be preserved for long periods of time. Decades of basic research on DNA hybridization thermodynamics[14–16] and kinetics[17,18] have resulted in a well-accepted model for predicting the dissociation rate constant of mechanism (1). At ≤30 °C, the expected half-life of a typical 25 nt DNA duplex is expected to be over 500 years, while at ≥50 °C, the dissociation is more than 8 orders of magnitude faster (Supplementary Note 1).

Based on our understanding of DNA strand displacement biophysics[18–21], mechanism (2) is the likely the dominant mechanism for truth marker rearrangement at lower temperatures. Previous experimental studies observed very slow but measurable strand displacement rate constants for the four-way strand displacement reaction required for mechanism (2), in the absence of single-stranded sticky ends or toeholds[20,21]. However, it is possible that the true rate constant of four-way strand displacement is essentially 0, and that the previously observed low-yield displacement reaction was due to a fraction of poorly synthesized DNA oligos[22,23]. Our experimental results and mathematical modeling suggest that the true rate constants should be no higher than $0.1\,\mathrm{M^{-1}\,s^{-1}}$ (Supplementary Note 1).

We first performed polyacrylamide gel electrophoresis experiments to experimentally test the viability of long-term information preservation in the metastable information solution (Fig. 2c). To facilitate accurate quantitation of different species in this experiment, here we used a truth marker with a 5′ FAM functionalization, a false marker with a 5′ ROX functionalization, and a false message that was longer than the true message. Lanes 5 and 6 show that the information solution remains metastable over the course of 1 week at 25°, with insignificant rearrangement of truth marker to bind false messages. In contrast, lanes 7 through 9 show that the information solution rapidly relaxes to a ratiometric equilibrium at elevated temperatures.

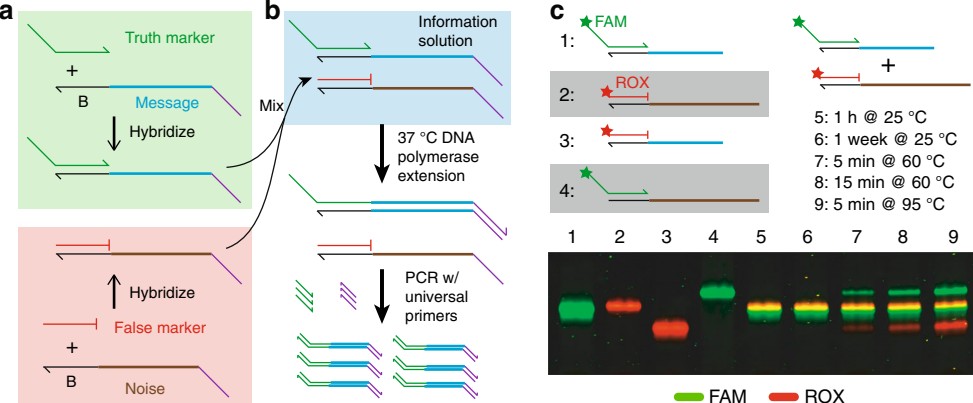

**Fig. 2 Metastable hybridization-based DNA information encoding. a** Preparing encoded messages. DNA oligos encoding the intended messages (cyan) are hybridized with truth marker oligos (green). In a separate solution, DNA oligos encoding noise messages (brown) are hybridized with false marker oligos (red). The two solutions are then mixed to form the information solution. **b** Reading the information solution. DNA polymerase and dNTPs are added to information solution, and the truth markers are extended to copy the intended messages. The false markers (red) have a three C3 functionalization and are unable to be extended. The mixture of extended truth markers is then amplified by PCR to generate a library encoding the true messages, which can then be read by NGS. **c** Experimental characterization of truth marker binding stability via polyacrylamide gel electrophoresis. Shown in the bottom panel is a composite of two images of the same gel scanned using different fluorescence filter sets, with the FAM image false colored in green, and the ROX image false colored in red. Lanes 1 and 2 show the intended message prehybridized to the truth marker and the noise DNA prehybridized to the false marker, respectively. Lanes 3 and 4 show the intended message prehybridized to the false marker and the noise DNA prehybridized to the truth marker. Lanes 5–9 show the mixture of the species in Lanes 1 and 2 incubated for different amounts of time at different temperatures. See Supplementary Note 2 for further analysis details.

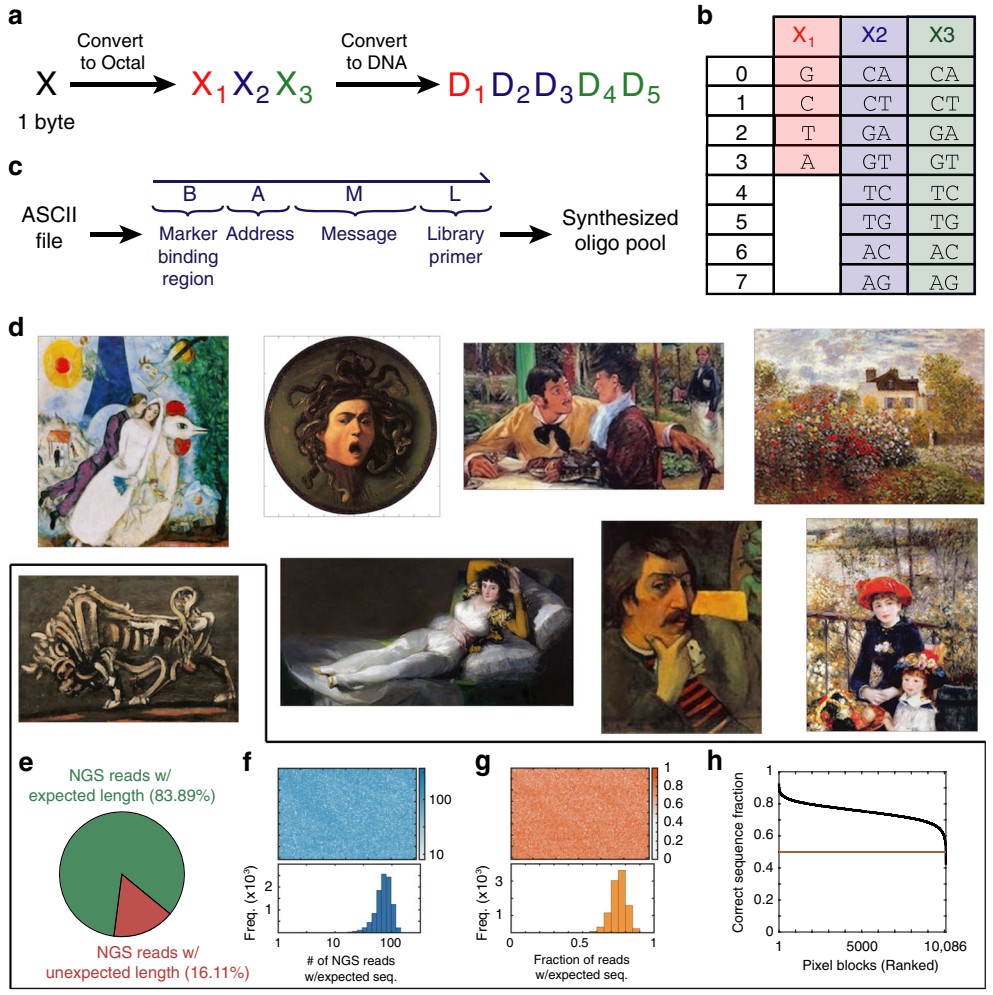

**Fig. 3 Encoding ASCII files as DNA.** The artworks used are low-resolution images freely available for reuse under the terms of a Creative Commons license or that are marked as being in the Public Domain. **a** We encode each byte as a word of 5 DNA nucleotides. The mapping is 80% efficient compared to the minimum 4 nt needed to encode 256 possible characters. **b** Mapping table. Importantly, this mapping restricts G/C content of DNA sequences to between 40% and 60%, and guarantees that there are no homopolymer stretches of more than 3 nt. **c** Each DNA oligonucleotide used for information storage can be abstracted as four domains. The B region is a 25 nt sequence common to all oligos, in which the truth marker and false marker can bind. The A region corresponds to the address of the message, relative to a file position, and is 10 nt in length. The M region corresponds to the message content and is 60 nt in length. The L region corresponds to a 25 nt library-specific primer sequence used for preamplification from chip-synthesized oligo pools; the L region is removed in the final oligos used for storage. **d** Bitmap images of eight pieces of artwork encoded as DNA. Displayed here are the reconstituted images based on the designed oligo pool synthesized by Twist Biosciences, read via NGS on an Illumina MiSeq. See Supplementary Note 4 for additional details. **e** Distribution of NGS reads that have the expected length. The 16.11% of NGS reads with unexpected lengths are typically shorter and are due to oligo synthesis truncations or sequencing error, and were excluded from further analysis. **f** Spatial distribution of sequencing depth. **g** Fraction of NGS reads mapping to the exact expected sequence for each pixel block, based on position. **h** Fraction of NGS reads mapping to the exact expected sequence for each pixel block, sorted by rank. 0.19% of the pixel blocks in this image had the intended NGS read sequence occupying below 50% of the reads at the address, indicating potential poor oligonucleotide synthesis quality.

**Encoding bitmap images in DNA.** Figure 3a shows our scheme for encoding standard ASCII data in DNA; each byte is converted into a 5 nt DNA string via the lookup table on Fig. 3b. This encoding is designed to facilitate DNA synthesis: all DNA sequences will have GC content between 40% and 60%, and no DNA sequence will have any nucleotide homopolymers of more than 3 (see Supplementary Note 3 for details). In our oligonucleotide pool of 93,894 distinct sequences, none triggered synthesis warnings, so we believe that the encoding successfully accomplishes its task of facilitating synthesis. Simultaneously, the encoding is 80% efficient from an information density perspective (encoding 8 bits with 10 bits), resulting in only a minor loss of information storage density.

For our demonstrations in this paper, we encoded bitmap images of famous artwork. Each pixel in the image corresponds to 3 bytes of information, one byte each for Red, Green, and Blue intensity, or 15 nt. Each DNA oligonucleotide (oligo) used for encoding information is 120 nt long, and comprises four distinct regions (Fig. 3c): the truth marker binding region B (25 nt), the address A (10 nt), the message M (60 nt), and the library primer L (25 nt). Each message M corresponds to a $2 \times 2$ block of pixels, and each address specifies the $X$ and $Y$ coordinates of the pixel block. The 93,894 DNA oligos in the pool were used to encode the eight images shown in Fig. 3d. Because the initially synthesized pool was very low in total quantity, we individually amplified different subpools via the image-specific library primer sequences (see Supplementary Note 4 for details).

Figure 3e–h show the quality of the reconstituted image for a single image, "The Bull", in the absence of false messages and false markers. Roughly 16% of all NGS reads indicated a message with incorrect length (Fig. 3e), and were discarded from further analysis. Of the remaining reads, there remain reads that cannot be decoded or may be decoded incorrectly: For example, if $X2 = 0$ in Fig. 3a, then a mutation of $D_2$ from C to T would result in an undecodable DNA sequence, but a mutation of $D_2$ from C to G would result in a decodeable DNA sequence with an error. Fig. 3f shows the spatial distribution of the number of NGS reads that perfectly match the expected NGS read (based on the original encoded image) at each pixel block, and Fig. 3g shows the fraction of reads at each pixel block corresponding to the expected NGS read. Roughly 0.19% of the pixel blocks had a correct message fraction <50%, suggesting that the majority of molecules for these sequences were incorrectly synthesized in the original Twist Biosciences oligo pool.

**Preservation of eight metastable information solutions for 65 days**. Finally, we experimentally tested the long-term stability of our metastable information solutions. We formulated eight separate information solutions, with the true messages corresponding to the images in Fig. 3d. In each information solution, the DNA oligos corresponding to the other seven images were prehybridized with the false markers, and added to the intended image prehybridized to the truth markers. Each information solution was incubated at 25 °C for 65 days, and then read out via NGS; the reconstructed images are displayed in Fig. 4a. These images were highly similar to images reconstructed after 1 h of incubation at 25 °C, and after 1 week of incubation at 25 °C

(Fig. 5a, b; see Supplementary Note 5 for further details and data analysis).

A separate aliquot of the eight different information solutions was heated to 95 °C for 5 min immediately after formulation, and then read out via NGS; those reconstructed images are displayed in Fig. 4b. Statistical analysis of these results (Fig. 5c, Supplementary Note 5) shows that it is not possible to confidently match any of the erased images to their originals at the $\alpha = 0.05$ significance level. These results are consistent with expectations since all eight erased information solutions have the same nominal concentrations of all DNA oligos at equilibrium.

In Fig. 4b, we plotted the decoded images based on the plurality of NGS reads at each pixel block address. The erased image contains recognizable pixels from each image because minor differences in standard free energy of truth marker binding to different DNA sequences do result in one DNA sequence in each pixel block occupying a consistent plurality of NGS reads. The different widths and heights of each of the eight images result in some pixel blocks being resembled the original image (e.g., the right side of Image 6) because there are no false messages in some of the pixel blocks. However, this is an artifact of our selection of images to encode; in actual use of our approach to encode information, it is easy to pad all images with random messages outside the regions of interest so that there are no obvious decoding of some pixel blocks.

Because of the observed variability in information solution preparation and NGS readout, we did not find a statistical difference in the fraction of pixels correctly restored across storage times of 1 h, 7 days, and 65 days at 25 °C (Fig. 5a, b). Based on the decrease in the correct pixel fraction of the 65 day storage relative to the original Twist Pool (99.21% vs. 99.81%), we

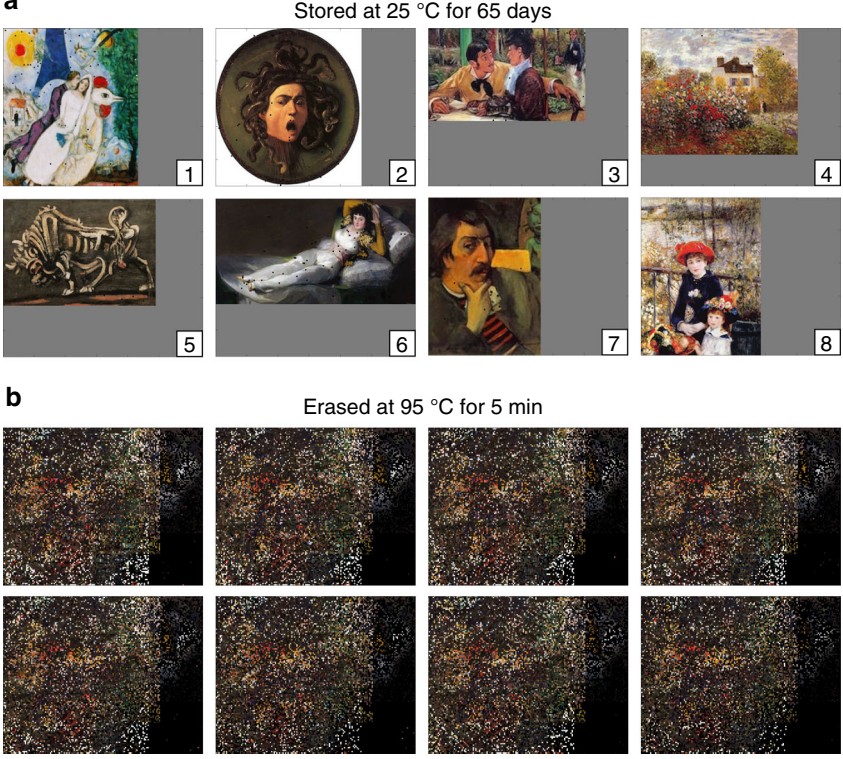

**a** Stored at 25 °C for 65 days

**b** Erased at 95 °C for 5 min

**Fig. 4 Demonstration of information storage and erasure on eight separate information solutions.** The artworks used are low-resolution images freely available for reuse under the terms of a Creative Commons license or that are marked as being in the Public Domain. **a** Reconstructed images from NGS reads after the information solutions were stored at 25 °C for 65 days. The bounding box here has dimensions of 320 pixels (horizontal) by 240 pixels (vertical); gray pixels indicate potential addresses outside of each image. **b** Reconstructed images from NGS reads after the information solutions were erased by incubating at 95 °C for 5 min.

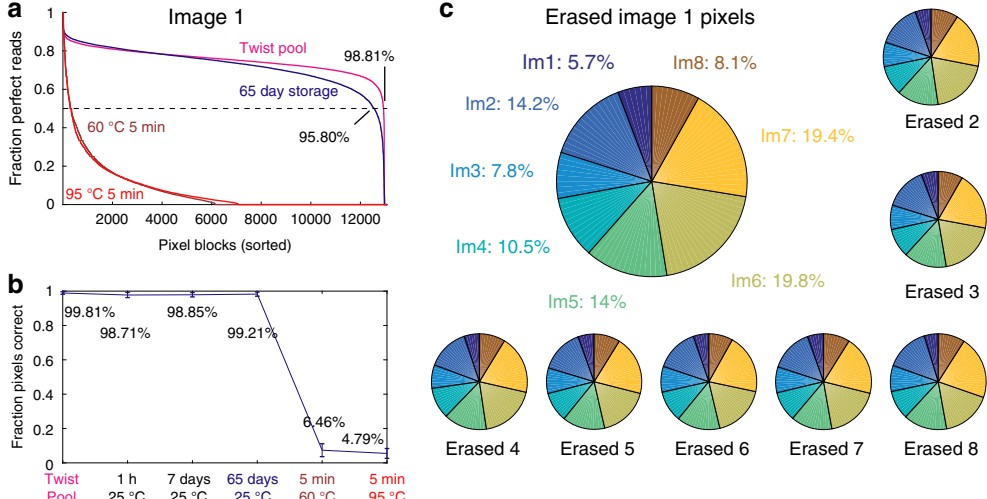

**Fig. 5 Analysis summary of image reconstruction and erasure completeness. a** Fraction of NGS reads mapped to each pixel block with expected sequence. The four different colored traces correspond to the different storage/erasure conditions. **b** Summary of fraction of pixels correctly restored for different storage/erasure conditions. Here, a pixel is considered to be correctly read if at least 50% of the decodable NGS reads at its address correspond to the expected sequence. The error bars show one standard deviation for the $N = 8$ separate restored images. **c** Distribution of best-match pixels for each erased image shown in Fig. 4b. For example, 14.2% of the erased Image 1 pixels have a consensus sequence that matches Image 2. A different fraction of pixels match best to each of the eight reference images, likely due to differences in initial concentrations of oligos from either the Twist pool or due to biased PCR amplification. All eight erased images have similar distributions of best-match pixels, suggesting that erasure is complete and images cannot be restored.

can lower-bound the half-life of the information to be $\frac{99.81\% - 50\%}{99.81\% - 99.21\%} \times 65$ days $= 5.4 \times 10^3$ days $\approx 15$ years. Storage at lower temperatures (e.g., 4 °C) likely would improve the information half-life by a factor of 100, based on experimental studies of the relative kinetics of DNA hybridization and strand displacement at different temperatures. Based on our experiments in which the information solution was incubated for 5 min at 60 °C and 5 min at 95 °C, we believe that a sufficiently high fraction the truth marker pairing information has been erased that the original information cannot be faithfully restored.

## Discussion

The chemical stability of DNA allows long-term information storage, but also prevents rapid and permanent erasure of information. Conventional methods to destroy DNA rely either on enzymes that require specific storage and buffer conditions, or harsh chemical/physical conditions such as ultraviolet light irradiation or temperatures over 200 °C. What is more, these methods often leave traces of DNA oligonucleotides, which makes data exposure possible. Here, we introduced the concept of using hybridization state metastability to allow durable DNA-based information storage with the ability to be rapidly, permanently, and easily erased via heating. Our experiments show that when the metastable information solution is stored for over 2 months at 25 °C, more than 99% of pixels (information) is correctly restored, suggesting an information half-life of at least 15 years. In contrast, 5 min at 95 °C is sufficient to completely erase the truth marker pairing, and render the encoded information unrecoverable.

In this work, we did not include any error correction methods or DNA sequences, in order to more clearly visualize the information preservation and erasure via graphical images. Standard error correction schemes such as Reed-Solomon encoding[24] or Fountain codes[4] can be easily integrated with our metastable hybridization-based encoding method, in order to allow robust information recovery from information solutions stored for

extended periods of time. Other encoding schemes, such as using degenerate/composite DNA letters[25,26], can also be easily combined with our method. Likewise, DNA file indexing approaches[5] can also be integrated.

The messages of the images are shown only for visualization purposes, and do not represent realistic information encoding schemes that would likely have higher K-complexity than bitmap images. In principle, error-correcting codes can be designed so that information is restorable from a very minute fraction of the original information, at the cost of low information and encoding inefficiency. However, we expect that for actual use the information solution will be designed to be easily erasable via heating, e.g., requiring 80% of original messages to be present for lossless restoration.

The metastability of the truth marker pairing is limited by the kinetics of DNA dissociation and double-stranded DNA strand displacement. Both processes are highly temperature sensitive, so we believe that storing the information solution at a low temperature (e.g., 4 °C) can maximize the half-life of the information solution. Our approach may not be compatible with frozen or lyophilized storage of DNA solutions, due to the poorly understood biophysics of Watson–Crick base pairs during freezing and vacuum-dry processes. On the other hand, treating the information solutions through formalin-fixing paraffin-embedding process may be an effective way to reduce storage/shipping requirements.

In principle, design of longer truth marker binding regions can also increase the durability of information storage, with no strict upper limit. Doing so, however, decreases the fraction of each DNA molecule's sequence that encodes useful information. For example, we currently devote 25 nt out of 120 nt (20.8%) of our sequence for hybridization truth encoding; lengthening the truth marker to 30 nt will increase the half-life of the information solution by a minimum of 44% and possibly as much as 100,000-fold (see Supplementary Note 1). Improvements in DNA oligonucleotide synthesis technologies to efficiently manufacture oligo, such as recent demonstrations in enzymatic DNA synthesis[27,28], may render this a feasible option in the future.

More generally, we believe that the current work presents an important qualitative advance to the field of DNA information storage, by leveraging hybridization states to encode information in a different dimension than sequence. The use of multi-stranded DNA molecules allows combinatorial information content far exceeding encoding using DNA sequence alone, in analogy to the diversity of molecules versus the diversity of atoms. Additionally, whereas atoms (DNA sequences) are relatively immutable and robust, the bonds that hold together molecules (multi-stranded DNA complexes) are more fragile, allowing programmable rearrangement of atoms into different molecules.

## Methods

**Gel demonstration of information solution storage and erasure**. The PAGE experiments used an Invitrogen Novex precast 10% acrylamide TBE gel with 12 wells (Catalog# EC62752BOX). The gel was run for 80 min at 100 V, at 25 °C in 1× TBE running buffer. The gel was scanned using a Typhoon FLA 9500 (G/E Healthcare) using 500 V on the photomultiplier tube, and a resolution of 50 μm. See Supplementary Note 2 for further analysis details. Unprocessed gel scans for FAM, ROX, and SYBR GOLD channels are included in Supplementary Fig. 2.

**Twist oligo pool amplification and analysis**. We encoded 8 bitmap images into DNA sequences using our encoding algorithm which is available in the Supplementary Software file. Oligo sequences are available in Supplementary Data 1. We ordered two chip-synthesized DNA oligonucleotide pools from Twist Bioscience. The first pool contains a total of 93,894 DNA oligonucleotides encoding eight separate bitmap image files. All oligonucleotides are 120 nucleotides long. Based on NGS results on the Twist pool to check oligo quality and concentrations, we designed and ordered a second repair pool which contains the 975 oligos in which the NGS library on the original pool showed <10 perfect reads each. After receiving the pools in dry (lyophilized) form, we added 1× Tris-EDTA (TE) buffer to formulate a stock with concentration 10 ng/μL. Then we diluted each pool 10,000-fold using 0.1× TE buffer with 0.1% Tween-20 to form a secondary stock.

Primers for amplifying different subpools of oligos (corresponding to the eight separate bitmap image files) were ordered from Integrated DNA Technologies. The forward primer is common to all subpools, binding to marker binding region (B). The reverse primer sequence is different for each subpool, based on the sequence of the unique library primer (L). The forward primer and reverse primers have corresponding Illumina Nextera adapter sequences at the 5′ end.

Five microliter of the oligo pool secondary stock were mixed with 5 μL of the forward primer (4 μM), 5 μL of the reverse primer (4 μM), 25 μL KAPA Hifi enzyme mix, and 10 μL MilliQ water in a 0.6 mL Eppendorf tube. This 50 μL mix was then amplified via PCR using the following thermocycling protocol: (1) 95 °C for 3 min, (2) 98 °C for 20 s, (3) 60 °C for 20 s, (4) 72 °C for 15 s, (5) repeat (2)–(4) for 24 times, (6) 72 °C for 1 min. (25 cycles of amplification in total). The 50 μL amplicon solution was then purified using Agencourt AMPure XP beads (90 μL, 1.6×) following manufacturer's specifications. DNA molecules were eluted by 0.1× TE buffer with 0.1% Tween-20. The purified products were then quantitated using a Qubit dsDNA Assay kit.

The concentration of each oligo for a subpool was calculated by dividing Qubit-inferred total DNA concentration by the number of total oligos in the subpool. The PCR amplified subpool of the first Twist pool was mixed with the PCR amplified subpool of the second Twist pool at a 1:2 oligo concentration ratio (Supplementary Fig. 1).

Then Index primers were appended using the Nextera XT kit and the KAPA Hifi enzyme mix following manufacturer's specifications to prepare NGS samples. Amplicons were purified using Agencourt AMPure XP beads (0.9×), and then quantitated using a Qubit dsDNA HS Assay kit and diluted to the recommended concentration suggested by Illumina for the MiSeq instrument. Purified amplicons were also subject to a quality control assay using a Bioanalyzer capillary electrophoresis assay (Agilent). PhiX DNA solution was spiked in to occupy 15% of all molecules, consistent with Illumina recommendations. This final library was then run on an Illumina Miseq instrument using a v3-150 cycle kit.

**Preparing and reading the information solution**. Primers for amplifying different subpools of oligos (corresponding to the eight separate bitmap image files) were ordered from Integrated DNA Technologies. The common forward primer binds to marker binding region (B), while unique reverse primer binds to library primer (L). The forward primer is phosphorylated at the 5′ end, and the reverse primers have three phosphorothioate DNA nucleotides at the 5′ end.

Five microliter of the oligo pool secondary stock was mixed with 5 μL of the forward primer (4 μM), 5 μL of the reverse primer (4 μM), 25 μL KAPA Hifi enzyme mix, and 10 μL MilliQ water in a 0.6 mL Eppendorf tube. This 50 μL mix was then amplified via PCR using the following thermocycling protocol: (1) 95 °C for 3 min, (2) 98 °C for 20 s, (3) 60 °C for 20 s, (4) 72 °C for 15 s, (5) repeat (2)–(4) for 24 times, (6) 72 °C for 1 min (25 cycles of amplification in total). The 50 μL

amplicon solution was then purified using Agencourt AMPure XP beads (90 μL, 1.8×) following manufacturer's specifications. Subsequently, 20 μL of the purified amplicon solution was mixed with 1 μL Lambda Exonuclease enzyme (New England Biolabs), 3 μL Lambda Exonuclease reaction buffer (10×), and 6 μL MilliQ water. The mixture was incubated at 37 °C for 30 min and then at 75 °C for 10 min, in order to digest phosphorylated DNA molecules (extended forward primers), but not the phosphorothioate-modified DNA molecules (extended reverse primers). The products of this reactions were then purified using an Oligo Clean & Concentrator kit (Zymo Research) according to manufacturer's specifications. DNA molecules were eluted by 0.1× TE buffer with 0.1% Tween-20. The purified products were then quantitated using a Qubit ssDNA Assay kit. The concentration of each oligo for a subpool was calculated by dividing Qubit-inferred total DNA concentration by the number of distinct oligo species in the subpool. Then the subpool amplified from first original pool and the subpool amplified from second repair pool are mixed at a concentration ratio of 1:2 (Supplementary Fig. 2).

Then we mixed each subpool with either truth marker oligos or false marker oligos to formulate an information solution. For DNA subpools intended to be information DNA molecules, the subpool was mixed with 0.5× relative quantity of truth marker oligos in 0.1× PBS buffer. For DNA subpools intended to be noise DNA molecules, the subpool was mixed with 2× relative quantity of false marker oligos in 0.1× PBS buffer. The solutions were individually thermally annealed, cooling from 95 to 25 °C over the course of 70 min, using an Eppendorf Personal Mastercycler. The two solutions are then mixed at room temperature (25 °C) to form the information solution. The information solution was then separated into two aliquots. One aliquot was directly moved to the next enzymatic step. Another was erased first, which was typically by heating to 95 °C for 5 min, using an Eppendorf Personal Mastercycler, and then moved to the next step.

To read both the original information solution and erased solution, we applied several enzymatic steps as follows. To a 4 μL volume of information solution, we added 2 μL of Klenow fragment DNA polymerase, 1.25 μL of 1 mM dNTP mixture, 2 μL NEB Buffer 2, and 10.75 μL MilliQ water. The mixture was then incubated at 37 °C for 1 h to extend the truth markers. Subsequently, the sample was diluted tenfold with 0.1× TE buffer with 0.1% Tween-20. To 2.5 μL of the diluted mix, we added 12.5 μL KAPA Hifi enzyme mix, 2.5 μL forward primer (4 μM), and 7.5 μL reverse primer mixture (4 μM). This 25 μL mix was amplified via PCR using the following thermocycling profile: (1) 95 °C for 3 min, (2) 98 °C for 20 s, (3) 60 °C for 20 s, (4) 72 °C for 15 s, (5) repeat (2)–(4) once, and (6) 72 °C for 1 min.

Index primers were appended using the Nextera XT kit and the KAPA Hifi enzyme mix following manufacturer's specifications. Amplicons were purified using Agencourt AMPure XP beads (0.9×), and then quantitated using a Qubit dsDNA HS Assay kit and diluted to the recommended concentration suggested by Illumina for the MiSeq instrument. Purified amplicons were also subject to a quality control assay using a Bioanalyzer capillary electrophoresis assay (Agilent). PhiX DNA solution was spiked in to occupy 15% of all molecules, consistent with Illumina recommendations. This final library was then run on an Illumina Miseq instrument using a v3-150 cycle kit.

The sequencing results were decoded to reconstruct the images using our decoding algorithm and further analyzed with our own codes, which are available in the Supplementary Note 7: Software/Code zip file.

**DNA degradation by other methods**. For comparison, we performed DNA degradation using UV and using DNAse enzyme in Supplementary Note 6. For UV irradiation experiments, we aliquoted 100 ng of human gDNA in 6 μL of 1× TE buffer and irradiated with 252 nm wavelength UV light (C.B.S. SCIENTIFIC P-036-202-SS-IL) for 2 h in a PCR box. During the irradiation, the DNA solution was in a 0.2 mL Axygen PCR tube. For DNAse digestion experiments, we aliquoted 100 ng of human gDNA and treat with DNAse I (New England Biolabs) following manufacturer's protocol. After digestion, the solution was buffer exchanged using the Zymo DNA clean and concentrator-5 kit.

To quantitate the amount of DNA degradation, we performed qPCR analysis with a custom human PCR primer set (forward primer: GTAGCCGCTTCTCTGTGA GTT, reverse primer: GGCGGGGGCTTCTCTG) intended to generate a 77 nt amplicon. 3 μL of each DNA sample was mixed with 3 μL of the forward primer (4 μM), 3 μL of the reverse primer (4 μM), 16 μL ITaq Universal SYBR Green Supermix, and 6 μL MilliQ water in a 0.6 mL Eppendorf tube. This 30 μL enzyme mix was then separated into triplicate and performed qPCR using the following protocol: (1) 95 °C for 3 min, (2) 95 °C for 10 s, (3) 60 °C for 30 s, (4) repeat (2)–(3) for 60 times.

**Reporting summary**. Further information on research design is available in the Nature Research Reporting Summary linked to this article.

## Data availability

All experimental data are summarized in the Supplementary information, and numerical data files are available upon request. Any additional relevant data are available upon reasonable request.

## Code availability

All codes are available in a Supplementary zip file.

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

## Acknowledgements
This work was funded by NHGRI grant R01HG008752 to D.Y.Z., and a Packard Fellowship to M.B. The authors thank Jianyi Nie for editorial assistance.

## Author contributions
J.K. conceived the project, designed and conducted the experiments, analyzed the data, and wrote the paper. J.H.B. conducted the experiments and analyzed the data. M.B. provided discussions and edited the paper. D.Y.Z. conceived the project, analyzed the data, and wrote the paper.

## Competing interests
D.Y.Z. is a co-founder and significant equity holder of Nuprobe Global and of Torus Biosystems. There is a patent (patent number: us 62/675362) pending on the methods for metastable hybridization-based information encoding in DNA. All other authors declare no competing interests.
