## [Peer Review File · Nature Communications]

Reviewers' Comments:

Reviewer #1:

Remarks to the Author:

In this manuscript the authors address the problem of erasing digital information that is encoded and stored using DNA molecules. They suggest an encoding scheme, based on basic mechanisms in molecular biology, which allows for the permanent erasure of the information simply by heating the DNA molecules. Increased temperature drives denaturing of double stranded DNA molecules. The resulting ssDNA molecules then randomly hybridize so that ~50% of the True message molecules are replaced with False message molecules. This step is supposed to render the information unreadable. The authors demonstrate the method by encoding 8 bitmap images. They demonstrate coding and decoding of the information as well as performing the proposed erasure process. They show, experimentally, that said proposed erasure process leads to a sample in which the original data is lost, at least when using naïve reading.

The problem addressed in this manuscript is an interesting aspect of DNA based storage systems that was not addressed by previous work. The paper is well written, the method is described very clearly and all experimental work presented in the manuscript is sound and valid. However, we have major concerns regarding the work, as follows.

1. In the introduction the authors refer to chemical alternatives for the destruction of the encoded data claiming that they require "significant time". This claim needs to be more concrete and properly supported. In addition, the authors fail to mention common molecular biology mechanisms for the destruction of DNA molecules such as nucleases. This point needs to be clarified as it serves as one of the main motivations for the development of the suggested method. In fact – the authors should provide even more strawman ideas for performing the same task and convince the reader that their approach is advantageous in this context.

2. Since the authors aim to achieve data security with the ability to erase the encoded data they should present a more careful analysis of 'information leakage scenarios'. The authors should explicitly state the assumption on possible 'attacks' that attempt to recover the data. This type of analysis is routinely done in cryptography and frameworks do exist that can allow for such type of evaluation here. Furthermore – recent literature even address the removal of records in a cryptographic framework, which may serve as a reference point here.

See, for example, doi: [10.1007/978-3-540-70583-3_42](https://doi.org/10.1007/978-3-540-70583-3_42)

3. In particular, the authors should analyze possible recovery procedures, potentially employed by the attacker to overcome the suggested erasure. The explanation given in [22] is very simplistic and holds only in the case where no information on the encoded data or the encoding scheme is available for the attacker. This is not the case in most scenarios (for example, in the demonstration on bitmap images, the spatial nature of the encoding can be exploited to come up with possible incremental reconstruction procedures). This is a major concern that needs to be addressed for the suggested method.

4. Following the previous comment, the authors don't address the encoding of different file types or the use of different encodings that might pose limitations on the method. For example, the authors state that other encoding, including fountain codes, may be combined with the suggested method. This may not be the case as fountain code has two properties that makes this combination problematic. (a) Fountain code has internal redundancy protecting it from dropout of some of the oligos. This poses limitations on the efficiency of the erasure. (b) Fountain code is sensitive to "false positive" oligos. The accidental inclusion of a False oligo in the decoding process, which can happen before erasure (increased temperature), may render the entire message unreadable.

5. The method imposes a significant overhead in terms of DNA synthesis as it requires doubling the library size to accommodate the False messages. This point is ignored in the manuscript and must be addressed.

In addition to that, the following details should also be addressed:

- The authors state that the erased images were "statistically identical". This term is repeated in Supplementary Sections S5. No statistical test is performed to support this claim. Nor is the term

rigorously defined in the context of its current usage in the manuscript.

- The description of the noise information included in the False oligos is not clear. What is the nature of the information included as noise? Is it the same for all pixels? Does it follow the same encoding rules like the True messages? This information is important to evaluate possible recovery attempts (attacks) of the erased data (as above).
- In the discussion section, there is a reference marked with "?" [21?]. You might want to double check this one.

Reviewer #2:

Remarks to the Author:

The paper by Jangwon Kim et al. describes a way of permanently deleting information stored in DNA. This is elegantly solved by mixing the original information with noise, and having the original data and noise information "marked" with by a corresponding hybridised oligo. Upon heating the DNA, the marking is lost, and noise and information can no longer be differentiated, leading in a loss of information.

While the temperature sensitivity is solved in a very elegant way, the applicability of the proposed technology is still very open. The authors may want to elaborate somewhat more on applications this technology may have.

Additionally, the authors should compare the proposed technology to other ways of deleting information from DNA (e.g. by reaction with ROS, UV irradiation, or by processing with significantly higher temperatures ($\gg 200^\circ\text{C}$), see e.g. Paunescu et al. Nat Protoc. 2013, and other DNA stability tests available in literature).

The manuscript is well written, concise and to the point.

Just to be on the safe side, the authors, and/or editor may want to ensure that no copyrights are infringed by printing copies of these famous artworks. (art copyright can be a very complicated topic).

Response to Reviewers' comments

Reviewer #1 (Remarks to the Author):

In this manuscript the authors address the problem of erasing digital information that is encoded and stored using DNA molecules. They suggest an encoding scheme, based on basic mechanisms in molecular biology, which allows for the permanent erasure of the information simply by heating the DNA molecules. Increased temperature drives denaturing of double stranded DNA molecules. The resulting ssDNA molecules then randomly hybridize so that ~50% of the True message molecules are replaced with False message molecules. This step is supposed to render the information unreadable. The authors demonstrate the method by encoding 8 bitmap images. They demonstrate coding and decoding of the information as well as performing the proposed erasure process. They show, experimentally, that said proposed erasure process leads to a sample in which the original data is lost, at least when using naïve reading.

The problem addressed in this manuscript is an interesting aspect of DNA based storage systems that was not addressed by previous work. The paper is well written, the method is described very clearly and all experimental work presented in the manuscript is sound and valid. However, we have major concerns regarding the work, as follows.

1. In the introduction the authors refer to chemical alternatives for the destruction of the encoded data claiming that they require "significant time". This claim needs to be more concrete and properly supported. In addition, the authors fail to mention common molecular biology mechanisms for the destruction of DNA molecules such as nucleases. This point needs to be clarified as it serves as one of the main motivations for the development of the suggested method. In fact – the authors should provide even more strawman ideas for performing the same task and convince the reader that their approach is advantageous in this context.

We have added new experiments on DNA degradation by UV and by nuclease treatment to the Supplementary Materials in a new Section S6. Based on these results, we have added the following text to the manuscript:

"Conventional methods to destroy DNA rely either on enzymes that require specific storage and buffer conditions~\cite{Dimo2013}, or harsh chemical/physical conditions such as ultraviolet light irradiation or temperatures over 200~\$^\circ\$C~\cite{Paunescu2013}. Because complete and irreversible destruction of DNA molecules encoding information requires a long time or highly specialized equipment (Supplementary Section~S6), these are not practical for routine use in decentralized settings."

2. Since the authors aim to achieve data security with the ability to erase the encoded data they should present a more careful analysis of 'information leakage scenarios'. The authors should explicitly state the assumption on possible 'attacks' that attempt to recover the data. This type of analysis is routinely done in cryptography and frameworks do exist that can allow for such type of evaluation here. Furthermore – recent literature even address the removal of records in a cryptographic framework, which may serve as a reference point here. See, for example, doi: 10.1007/978-3-540-70583-3_42

Once truth marker and false marker binding is randomized through heating to 60 °C or 95 °C, the system has reached equilibrium and the encoded information is irrecoverable by any means. All cryptographic methods to recover information from lossy or damaged files require a minimal "core fraction" of correct information to be retained (e.g. between 80% and 85%), which is much higher than what we observe after 5 minutes of incubation to 95 °C.

We do not believe that attacks on the information solution prior to erasure (e.g. intercepting the solution before the owner has a chance to erase via heating) are not relevant to the current manuscript.

3. In particular, the authors should analyze possible recovery procedures, potentially employed by the attacker to overcome the suggested erasure. The explanation given in [22] is very simplistic and holds only in the case where no information on the encoded data or the encoding scheme is available for the attacker. This is not the case in most scenarios (for example, in the demonstration on bitmap images, the spatial nature of the encoding can be exploited to come up with possible incremental reconstruction procedures). This is a major concern that needs to be addressed for the suggested method.

We have added the following text to the manuscript:

“A brute force attack (that the reviewer suggests) would have $(M+1)^N$ possible messages, where M is the number of false messages at each pixel block, and N is the number of pixel blocks. For $M = 7$ and $N = 100$, the number of possible encoded messages is $2 \cdot 10^{90}$, far more than the number of atoms in the universe (estimated to be 10^{80}). Furthermore, there would be no way to ascertain which of the $2 \cdot 10^{90}$ images could be correct. This is one of the mathematical problems that is provably NP-Hard even if there was a way of confirming whether a potential reconstructed image is correct.”

We also wish to point out that the bitmap nature of the images is present only to help the reader visually interpret and estimate the erasure completeness. Our approach is not constrained to 1:1 encoding of bitmap images with long-range spatial interactions.

4. Following the previous comment, the authors don't address the encoding of different file types or the use of different encodings that might pose limitations on the method. For example, the authors state that other encoding, including fountain codes, may be combined with the suggested method. This may not be the case as fountain code has two properties that makes this combination problematic. (a) Fountain code has internal redundancy protecting it from dropout of some of the oligos. This poses limitations on the efficiency of the erasure. (b) Fountain code is sensitive to "false positive" oligos. The accidental inclusion of a False oligo in the decoding process, which can happen before erasure (increased temperature), may render the entire message unreadable.

Internal redundancy protection will not protect against our method of erasure, as the limit of recovery is about 80% of original information for Fountain codes. To put it in another way, our heating-based erasure completely takes the system to equilibrium in 5

minutes, erasing all 20,000 pixel blocks, but at least 16,000 pixel blocks need to be recovered for Fountain codes to restore information.

As the purpose of this paper is to introduce a new method to encode erasable information, we would prefer not to delve deeply into specific limitations of Fountain codes such as false positives.

5. The method imposes a significant overhead in terms of DNA synthesis as it requires doubling the library size to accommodate the False messages. This point is ignored in the manuscript and must be addressed.

We have added additional text to the manuscript describing the potentially increased synthesis cost of the false messages, though with chip-based synthesis the cost of doubling the oligos is far less than doubling the initial cost.

We also wish to point out that any information-bearing DNA molecules can be either true oligos or false oligos. Once it is bound to the truth marker, a DNA molecule becomes a true oligo, while it becomes a false oligo if bound to the false marker. Thus, there need not be extra cost associated with synthesizing oligos for noise; noise can be oligos from encoding any other message.

In addition to that, the following details should also be addressed:

- The authors state that the erased images were “statistically identical”. This term is repeated in Supplementary Sections S5. No statistical test is performed to support this claim. Nor is the term rigorously defined in the context of its current usage in the manuscript.*

We thank the reviewer for alerting us to our lack of rigor on statistics. We have added the following text to the manuscript: "Statistical analysis of these results (Fig.~5c, Supplementary Section~S5) shows that it is not possible to confidently match any of the erased images to their originals at the $\alpha = 0.05$ significance level." Additionally, we have added statistical analysis in Supplementary Section~S5 to support our argument.

- The description of the noise information included in the False oligos is not clear. What is the nature of the information included as noise? Is it the same for all pixels? Does it follow the same encoding rules like the True messages? This information is important to evaluate possible recovery attempts (attacks) of the erased data (as above).*

We have added the following to our text:

“Importantly, any DNA molecule with sequence satisfying our expected format can be used either as a true message or a false message, and furthermore may be used as both a true message for one information solution and as a false message for a different information solution.”

- In the discussion section, there is a reference marked with “?” [21?]. You might want to double check this one.*

Thanks, we've fixed the reference link.

Reviewer #2 (Remarks to the Author):

The paper by Jangwon Kim et al. describes a way of permanently deleting information stored in DNA. This is elegantly solved by mixing the original information with noise, and having the original data and noise information "marked" with by a corresponding hybridized oligo. Upon heating the DNA, the marking is lost, and noise and information can no longer be differentiated, leading in a loss of information.

While the temperature sensitivity is solved in a very elegant way, the applicability of the proposed technology is still very open. The authors may want to elaborate somewhat more on applications this technology may have.

We mentioned in the introduction that this method may be useful for highly confidential information such as personal genomic information or military secrets. We feel that, given the early stage of development of this technology, further speculation on applications would not be too meaningful.

Additionally, the authors should compare the proposed technology to other ways of deleting information from DNA (e.g. by reaction with ROS, UV irradiation, or by processing with significantly higher temperatures (>>200°C), see e.g. Paunescu et al. Nat Protoc. 2013, and other DNA stability tests available in literature).

We have added new experiments on DNA degradation by UV and by nuclease treatment to the Supplementary Materials in a new Section S6. Based on these results, we have added the following text to the manuscript:

"Conventional methods to destroy DNA rely either on enzymes that require specific storage and buffer conditions~\cite{Dimo2013}, or harsh chemical/physical conditions such as ultraviolet light irradiation or temperatures over 200~\$^\circ\$C~\cite{Paunescu2013}. Because complete and irreversible destruction of DNA molecules encoding information requires a long time or highly specialized equipment (Supplementary Section~S6), these are not practical for routine use in decentralized settings."

The manuscript is well written, concise and to the point.

Just to be on the safe side, the authors, and/or editor may want to ensure that no copyrights are infringed by printing copies of these famous artworks. (art copyright can be a very complicated topic).

These artworks are low-resolution images freely available for reuse under the terms of a Creative Commons license or that are marked as being in the Public Domain. We have included text on artwork use in the "Additional Information" at the end of the manuscript.

Reviewers' Comments:

Reviewer #1:

Remarks to the Author:

We have carefully read the revised manuscript together with the authors' responses to all comments. The authors have only partially addressed the issues brought up in the first round of review. We feel that by increasing the scientific accuracy of the writing and by directly and rigorously addressing certain technical aspects of the work the manuscript will greatly improve and will then be appropriate for publication.

Specifically:

1. The authors addressed our comment about comparing to other potential approaches. The point about specialty reagents is well taken and the new data reported in the DNA degradation by UV and by nuclease treatment is relevant and supporting the advantage of the proposed method. However, the question about heating to higher temperatures remains. How about putting the test tube in the home oven at 250C? Karni et al. (2013, DOI: 10.1089/dna.2013.2056) suggest that not much DNA will remain. The authors should address this in the manuscript. In particular, they should clearly define the advantages of the suggested approach vs the home oven. The advantage vs specialty reagents is well clarified.

2. With regards to the characterization of the information leakage of the suggested erasure method, the authors did not address our comments to our satisfaction.

a. Looking at the pictures in Figure 4b, it is evident that the elliptic frame (BTW – how are the images related to the image numbers used eg in FigS5-7?) is visible in all information solutions together with some other features from the original files. This means that some information of the original stored data remains accessible after the erasure. This should be quantified and addressed.

b. Moreover, the fact that this elliptic shape is visible in all the solutions means that using other images as the “false messages” (or decoy) results in exposure of information from other files to the (possibly malicious) reader of any erased original file. This may be disturbing from the point of view of potential users that may find sensitive information about their data exposed to others.

c. As to the question of how to quantify the security of the suggested method. We did not mean brute force attack and we recommend removing the added text that addresses this. Moreover, all reference to the full space of all $(M+1)^N$ possible messages is irrelevant since the true space of reasonable images is much smaller.

d. The authors should discuss possible reconstruction methods that uses the limited information that leaked from the erasure. Most likely the original images will not be recoverable to their full extent. However, as mentioned above, some information remains and may be used for a partial recovery of the messages. There is some literature on this topic. Even if the issue can not be quantitatively addressed it should still be emphasized in the discussion.

3. The t-test performed to assess the leakage is problematic in our view.

a. Getting an insufficient p-value means you cannot reject the null hypothesis. Nonetheless - FigS5-7 is informative.

b. If you perform t-test of the number of pixels remaining as compared to a distribution of the number of common pixels a random image would have with Image 5, say, then you would get an extremely significant t-test. This is because much of the information DOES remain after erasure.

c. One possible alternative approach to quantifying the similarity between the erased images would be to show that the ranking of the original pixel block over the 8 possible pixel blocks is uniformly distributed when examining all the output pixels in the deleted solutions. That is, when observing $8*100=800$ ranking vectors (of length 8 each), the rank of the original pixel, R , should satisfy $R \sim \text{Unif}(1,8)$.

4. The authors may want to explore the possibility of combining their approach with newer coding approaches for DNA based storage. This include the use of native DNA (Tabatabaei et al, DOI: 10.1101/672394) and the use of degenerate/composite DNA letters (Choi et al, DOI: 10.1038/s41598-019-43105-w; Anavy et al, DOI: 10.1038/s41587-019-0240-x). Specifically, combining the proposed erasure approach with composite DNA letters may reduce the leaked information to the bare minimum since the encoding is based on the composition in each position rather than using the majority vote to select the base.

In summary, all this is not necessarily an obstacle to the publication of the manuscript. We do feel, however, that these issues should be addressed and emphasized as part of the manuscript. Doing so will greatly improve the quality of the manuscript, making it a better fit for publication.

Reviewer #2:

Remarks to the Author:

In revising their manuscript, the authors have improved the manuscript, but many of the questions / comments of the referees in the first round of review were not fully answered, and did not result in an improvement of the manuscript. Obviously these input is outside the capacity and scope of the authors. Still, I believe that the manuscript should be published, especially if the referee reports are published together with the manuscript (as routinely done by Nat Commun). However, the two points below are false statements, and should be corrected prior to acceptance.

1. decay of DNA with other methods: The authors write that significant infrastructure is required to destroy DNA with other methods. This is just untrue, and also reflected in the supporting information Figure S6_1: just adding an enzyme, and adjusting the buffer (this can also be done by addition of buffer molecules and does not require buffer replacement) is sufficient to completely destroy the DNA. Also one could think of many other chemical methods for destroying the DNA, involving hydrogen peroxide, heavy metals, strong oxidizing agents (e.g. potassium permanganate), and even household bleach. Chemically DNA is a sensitive molecule, and destroying it is not difficult, even if only using standard chemicals. There are also commercial chemicals developed specifically for the purpose of degrading DNA (e.g. DNAzap), and these are very easy to use, and don't require any infrastructure.

In short: there are very rapid and easy chemical and biological ways to destroy information stored in DNA, they are just not as elegant as the method described in the manuscript. The text in the introduction should reflect this.

2. it is untrue that fountain codes require 80% of the DNA sequences to be present to successfully decode. The amount of redundancy can be chosen at will during encoding. This can be chosen so that only a few percent of the DNA sequences are required for decoding.

Reviewers' comments:

Reviewer #1 (Remarks to the Author):

We have carefully read the revised manuscript together with the authors' responses to all comments. The authors have only partially addressed the issues brought up in the first round of review. We feel that by increasing the scientific accuracy of the writing and by directly and rigorously addressing certain technical aspects of the work the manuscript will greatly improve and will then be appropriate for publication.

Specifically:

1. The authors addressed our comment about comparing to other potential approaches. The point about specialty reagents is well taken and the new data reported in the DNA degradation by UV and by nuclease treatment is relevant and supporting the advantage of the proposed method.

However, the question about heating to higher temperatures remains. How about putting the test tube in the home oven at 250C? Karni et al. (2013, DOI: 10.1089/dna.2013.2056) suggest that not much DNA will remain. The authors should address this in the manuscript. In particular, they should clearly define the advantages of the suggested approach vs the home oven. The advantage vs specialty reagents is well clarified.

We thank the reviewer for pointing out these alternative methods for DNA destruction, and have mentioned these in our revised manuscript. This also helped us clarify the unique advantage of our approach, that we can achieve rapid erasure of information in decentralized settings, e.g. during physical transport of the encoded information solution from the sender to a receiver.

2. With regards to the characterization of the information leakage of the suggested erasure method, the authors did not address our comments to our satisfaction.

a. Looking at the pictures in Figure 4b, it is evident that the elliptic frame (BTW – how are the images related to the image numbers used eg in FigS5-7?) is visible in all information solutions together with some other features from the original files. This means that some information of the original stored data remains accessible after the erasure. This should be quantified and addressed.

We thank the reviewer for pointing out the potential confusion. We have added the following clarification text: "The different widths and heights of each of the 8 images results in some pixel blocks being resembling the original image (e.g. the right side of Image 6) because there are no false messages in some of the pixel blocks. However, this is an artifact of our selection of images to encode; in actual use of our approach to encode information, it is easy to pad all images with random messages outside the regions of interest so that there are no obvious decoding of some pixel blocks."

b. Moreover, the fact that this elliptic shape is visible in all the solutions means that using other images as the “false messages” (or decoy) results in exposure of information from other files to the (possibly malicious) reader of any erased original file. This may be disturbing from the point of view of potential users that may find sensitive information about their data exposed to others.

We have added text to clarify that fractional remnant information does not represent a security risk: "Erasure of information, in our context, does not require destroying the potential messages by chemical degradation of DNA, but rather relies on the randomization of true/false flags encoded in hybridization state. Given a DNA pool with N addresses and M potential messages at each address, there are M^N possible files; the hybridization states of the $N \cdot M$ DNA oligos inform which one of the M^N messages is the intended message. Once hybridization state information is lost, it is impossible to identify the original message among all the possibilities."

See also below regarding our response to low fraction of remaining information.

c. As to the question of how to quantify the security of the suggested method. We did not mean brute force attack and we recommend removing the added text that addresses this. Moreover, all reference to the full space of all $(M+1)^N$ possible messages is irrelevant since the true space of reasonable images is much smaller.

We disagree with this comment, the space of "reasonable" files should in fact be $(M+1)^N$, because we envision that most users would use a compressed representation of information (e.g. zipfile) with high K -complexity. For this paper, we only showed bitmap encoding because it is easy for visualize.

We have added the following clarification text: " Here we use the bitmap artwork images to demonstrate our approach in a way that is easy to visualize; however, we do not require any long-range pattern in the underlying data, and we imagine that practical uses of this approach would be on compressed information files with high K -complexity."

d. The authors should discuss possible reconstruction methods that uses the limited information that leaked from the erasure. Most likely the original images will not be recoverable to their full extent. However, as mentioned above, some information remains and may be used for a partial recovery of the messages. There is some literature on this topic. Even if the issue can not be quantitatively addressed it should still be emphasized in the discussion.

We feel that we need to clarify the distinction between setting up a message where the intention is to be able to recover from minute traces of information (e.g. by encoding with high redundancy) vs. when we have an encoded message in which we would like

to enable rapid permanent erasure (e.g. those that require 80+% of original bits to enable lossless recovery). Paired with our commentary previously about encoding high K-complexity compressed files, we do not see significant risk of recovering the information after heat erasure.

We have added the following text: "In principle, error-correcting codes can be designed so that information is restorable from a very minute fraction of the original information, at the cost of low information content and encoding inefficiency. However, we expect that for actual use when the information solution will be designed to easily erasable via heating, e.g. requiring 80% of original messages to be present for lossless restoration."

3. The t-test performed to assess the leakage is problematic in our view.
a. Getting an insufficient p-value means you cannot reject the null hypothesis. Nonetheless - FigS5-7 is informative.

We consider the p-values shown in Fig. S5-7 a very strong result, because we have already down-selected from the 8^{76800} to simply 8 potential images. In a real use setting, all 8^{76800} potentially messages would be viewed as equally likely.

b. If you perform t-test of the number of pixels remaining as compared to a distribution of the number of common pixels a random image would have with Image 5, say, then you would get an extremely significant t-test. This is because much of the information DOES remain after erasure.

As explained previously, it is one thing to recover a message when you know a-priori that there are only 8 possible messages. In practice, there are 8^{76800} possible messages, and we do not realistically think recovering from a minute amount of remaining information is possible.

c. One possible alternative approach to quantifying the similarity between the erased images would be to show that the ranking of the original pixel block over the 8 possible pixel blocks is uniformly distributed when examining all the output pixels in the deleted solutions. That is, when observing $8 \times 100 = 800$ ranking vectors (of length 8 each), the rank of the original pixel, R , should satisfy $R \sim \text{Unif}(1,8)$.

As described previously, because all of the images encoded have different dimensions, there are certain pixel blocks that do not have corresponding false messages, so the rank would the correct pixel would be 1. This is not an intrinsic limitation of the system, but rather an artifact of the images we selected to encode.

We believe that Fig. S5-7 already sufficiently shows the difficulty of recovering the images. If the reviewer is interested, we would be happy to provide the NGS results in blinded format against the 8 erased images, and invite the reviewer to match each of these to the 8 original images.

4. The authors may want to explore the possibility of combining their approach with

newer coding approaches for DNA based storage. This include the use of native DNA (Tabatabaei et al, DOI: 10.1101/672394) and the use of degenerate/composite DNA letters (Choi et al, DOI: 10.1038/s41598-019-43105-w; Anavy et al, DOI: 10.1038/s41587-019-0240-x). Specifically, combining the proposed erasure approach with composite DNA letters may reduce the leaked information to the bare minimum since the encoding is based on the composition in each position rather than using the majority vote to select the base.

We added the following text to the manuscript: "Other encoding schemes, such as using degenerate/composite DNA letters, can also be easily combined with our method." We believe that it's beyond the scope of the current paper to redesign several more panels to show a relatively straightforward combination of our erasure approach with published encoding approach.

In summary, all this is not necessarily an obstacle to the publication of the manuscript. We do feel, however, that these issues should be addressed and emphasized as part of the manuscript. Doing so will greatly improve the quality of the manuscript, making it a better fit for publication.

Reviewer #1 Additional Comments:

"1. With respect to alternative approaches - how about flushing the DNA solution down the sink? True - some detergent needs to be used in order to guarantee minimal residue but this is still very simple ...

We thank the reviewer for pointing out this alternative method for DNA destruction. This also helped us clarify the unique advantage of our approach, that we can achieve rapid erasure of information in decentralized settings, e.g. during physical transport of the encoded information solution from the sender to a receiver.

2. With respect to image reconstruction - the following is an example of many papers describe approaches that require very small fraction of remaining pixels:

https://www.researchgate.net/publication/257731990_A_fast_novel_algorithm_for_salt_and_pepper_image_noise_cancellation_using_cardinal_B-splines

The images erased by the authors will be easily recovered by such approaches."

Again, the messages of the image are shown only for visualization purposes. The paper mentioned by the reviewer will be unable to losslessly restore an image from 7% correct pixel blocks. Note that we have added the following text: "In principle, error-correcting codes can be designed so that information is restorable from a very minute fraction of the original information, at the cost of low information content and encoding inefficiency. However, we expect that for actual use when the information solution will be designed to easily erasable via heating, e.g. requiring 80% of original messages to be present for lossless restoration."

Reviewer #2 (Remarks to the Author):

In revising their manuscript, the authors have improved the manuscript, but many of the questions / comments of the referees in the first round of review were not fully answered, and did not result in an improvement of the manuscript. Obviously these input is outside the capacity and scope of the authors. Still, I believe that the manuscript should be published, especially if the referee reports are published together with the manuscript (as routinely done by Nat Commun). However, the two points below are false statements, and should be corrected prior to acceptance.

1. decay of DNA with other methods: The authors write that significant infrastructure is required to destroy DNA with other methods. This is just untrue, and also reflected in the supporting information Figure S6_1: just adding an enzyme, and adjusting the buffer (this can also be done by addition of buffer molecules and does not require buffer replacement) is sufficient to completely destroy the DNA. Also one could think of many other chemical methods for destroying the DNA, involving hydrogen peroxide, heavy metals, strong oxidizing agents (e.g. potassium permanganate), and even household bleach. Chemically DNA is a sensitive molecule, and destroying it is not difficult, even if only using standard chemicals. There are also commercial chemicals developed specifically for the purpose of degrading DNA (e.g. DNAzap), and these are very easy to use, and don't require any infrastructure.

In short: there are very rapid and easy chemical and biological ways to destroy information stored in DNA, they are just not as elegant as the method described in the manuscript. The text in the introduction should reflect this.

We thank the reviewer for pointing out these alternative methods for DNA destruction, and have mentioned these in our revised manuscript. This also helped us clarify the unique advantage of our approach, that we can achieve rapid erasure of information in decentralized settings, e.g. during physical transport of the encoded information solution from the sender to a receiver.

2. it is untrue that fountain codes require 80% of the DNA sequences to be present to successfully decode. The amount of redundancy can be chosen at will during encoding. This can be chosen so that only a few percent of the DNA sequences are required for decoding.

We feel that we need to clarify the distinction between setting up a message where the intention is to be able to recover from minute traces of information (e.g. by encoding with high redundancy) vs. when we have an encoded message in which we would like to enable rapid permanent erasure (e.g. those that require 80+% of original bits to enable lossless recovery). Paired with our commentary previously about encoding high K-complexity compressed files, we do not see significant risk of recovering the information after heat erasure.

We have added the following text: "In principle, error-correcting codes can be designed so that information is restorable from a very minute fraction of the original information, at the cost of low information content and encoding inefficiency. However, we expect that

for actual use when the information solution will be designed to easily erasable via heating, e.g. requiring 80% of original messages to be present for lossless restoration."

Reviewers' Comments:

Reviewer #1:

Remarks to the Author:

Reviewers' comments:

Reviewer #1 (Remarks to the Author):

We have carefully read the revised manuscript together with the authors' responses to all comments. The authors have only partially addressed the issues brought up in the first round of review. We feel that by increasing the scientific accuracy of the writing and by directly and rigorously addressing certain technical aspects of the work the manuscript will greatly improve and will then be appropriate for publication. Specifically:

1. The authors addressed our comment about comparing to other potential approaches. The point about specialty reagents is well taken and the new data reported in the DNA degradation by UV and by nuclease treatment is relevant and supporting the advantage of the proposed method.

However, the question about heating to higher temperatures remains. How about putting the test tube in the home oven at 250C? Karni et al. (2013, DOI: 10.1089/dna.2013.2056) suggest that not much DNA will remain. The authors should address this in the manuscript. In particular, they should clearly define the advantages of the suggested approach vs the home oven. The advantage vs specialty reagents is well clarified.

We thank the reviewer for pointing out these alternative methods for DNA destruction, and have mentioned these in our revised manuscript. This also helped us clarify the unique advantage of our approach, that we can achieve rapid erasure of information in decentralized settings, e.g. during physical transport of the encoded information solution from the sender to a receiver.

>> OK. I think that all language should reflect this then. State in the abstract that many possible approaches will destroy the DNA. Including trivial actions such as heating to 250C or flushing in the sink w detergents, as well as vchemical approaches such as pointed out by Reviewer #2. The intro should also mention some of these with references. Then the situations where the methods proposed are useful should be pointed out.

2. With regards to the characterization of the information leakage of the suggested erasure method, the authors did not address our comments to our satisfaction.

a. Looking at the pictures in Figure 4b, it is evident that the elliptic frame (BTW – how are the images related to the image numbers used eg in FigS5-7?) is visible in all information solutions together with some other features from the original files. This means that some information of the original stored data remains accessible after the erasure. This should be quantified and addressed.

We thank the reviewer for pointing out the potential confusion. We have added the following clarification text: "The different widths and heights of each of the 8 images

results in some pixel blocks being resembling the original image (e.g. the right side of Image 6) because there are no false messages in some of the pixel blocks. However, this is an artifact of our selection of images to encode; in actual use of our approach to encode information, it is easy to pad all images with random messages outside the regions of interest so that there are no obvious decoding of some pixel blocks."

>> I accept the response and the text.

b. Moreover, the fact that this elliptic shape is visible in all the solutions means that using other images as the "false messages" (or decoy) results in exposure of information from other files to the (possibly malicious) reader of any erased original file. This may be disturbing from the point of view of potential users that may find sensitive information about their data exposed to others.

We have added text to clarify that fractional remnant information does not represent a security risk: "Erasure of information, in our context, does not require destroying the potential messages by chemical degradation of DNA, but rather relies on the randomization of true/false flags encoded in hybridization state. Given a DNA pool with N addresses and M potential messages at each address, there are M^N possible files; the hybridization states of the $N \cdot M$ DNA oligos inform which one of the M^N messages is the intended message. Once hybridization state information is lost, it is impossible to identify the original message among all the possibilities."

See also below regarding our response to low fraction of remaining information.

c. As to the question of how to quantify the security of the suggested method. We did not mean brute force attack and we recommend removing the added text that addresses this. Moreover, all reference to the full space of all $(M+1)^N$ possible messages is irrelevant since the true space of reasonable images is much smaller.

We disagree with this comment, the space of "reasonable" files should in fact be $(M+1)^N$, because we envision that most users would use a compressed representation of information (e.g. zipfile) with high K -complexity. For this paper, we only showed bitmap encoding because it is easy for visualize.

We have added the following clarification text: " Here we use the bitmap artwork images to demonstrate our approach in a way that is easy to visualize; however, we do not require any long-range pattern in the underlying data, and we imagine that practical uses of this approach would be on compressed information files with high K -complexity."

d. The authors should discuss possible reconstruction methods that uses the limited information that leaked from the erasure. Most likely the original images will not be recoverable to their full extent. However, as mentioned above, some information

remains and may be used for a partial recovery of the messages. There is some literature on this topic. Even if the issue can not be quantitatively addressed it should still be emphasized in the discussion.

We feel that we need to clarify the distinction between setting up a message where the intention is to be able to recover from minute traces of information (e.g. by encoding with high redundancy) vs. when we have an encoded message in which we would like to enable rapid permanent erasure (e.g. those that require 80+% of original bits to enable lossless recovery). Paired with our commentary previously about encoding high K-complexity compressed files, we do not see significant risk of recovering the information after heat erasure.

We have added the following text: "In principle, error-correcting codes can be designed so that information is restorable from a very minute fraction of the original information, at the cost of low information content and encoding inefficiency. However, we expect that for actual use when the information solution will be designed to easily erasable via heating, e.g. requiring 80% of original messages to be present for lossless restoration."

>> OK. See in other comments

3. The t-test performed to assess the leakage is problematic in our view.
a. Getting an insufficient p-value means you cannot reject the null hypothesis. Nonetheless - FigS5-7 is informative.

We consider the p-values shown in Fig. S5-7 a very strong result, because we have already down-selected from the 8^{76800} to simply 8 potential images. In a real use setting, all 8^{76800} potentially messages would be viewed as equally likely.

>> An insufficient p-value does not mean that you accept the null. It means you can not reject it.

b. If you perform t-test of the number of pixels remaining as compared to a distribution of the number of common pixels a random image would have with Image 5, say, then you would get an extremely significant t-test. This is because much of the information DOES remain after erasure.

As explained previously, it is one thing to recover a message when you know a-priori that there are only 8 possible messages. In practice, there are 8^{76800} possible messages, and we do not realistically think recovering from a minute amount of remaining information is possible.

>> For images this is not true. We can recover from very little information left. There is an issue w the use of 8^{76800} as the number of potential messages. Of course, a compressed image will be much more difficult to recover. But this is not what the authors are demonstrating. The point of showing all the images is a bit mute when its actually not what is meant. Analysis of the compressed case should be added and the

images should be less emphasized. An indication of the fact that they are only there for illustration purposes (which the authors state above: " Here we use the bitmap artwork images to demonstrate our approach in a way that is easy to visualize; however, we do not require any long-range pattern in the underlying data, and we imagine that practical uses of this approach would be on compressed information files with high K-complexity.") should be added. Furthermore - I do accept the argument re compressed messages but this argument should be made as a main argument. All arguments based on, or illustrated by, the images should become secondary. Otherwise the manuscript risks serious criticism of some claims that can not be applied to the images.

c. One possible alternative approach to quantifying the similarity between the erased images would be to show that the ranking of the original pixel block over the 8 possible pixel blocks is uniformly distributed when examining all the output pixels in the deleted solutions. That is, when observing $8 \times 100 = 800$ ranking vectors (of length 8 each), the rank of the original pixel, R , should satisfy $R \sim \text{Unif}(1,8)$.

As described previously, because all of the images encoded have different dimensions, there are certain pixel blocks that do not have corresponding false messages, so the rank of the correct pixel would be 1. This is not an intrinsic limitation of the system, but rather an artifact of the images we selected to encode.

We believe that Fig. S5-7 already sufficiently shows the difficulty of recovering the images. If the reviewer is interested, we would be happy to provide the NGS results in blinded format against the 8 erased images, and invite the reviewer to match each of these to the 8 original images.

>> Thanks for the invitation. The authors may want to propose a student project where the students will be asked to reconstruct Image 7 from the pixels provided. (as a side remark – the mapping btwn the images in F4 to the numbers in F5 is not clear to me). The hypothetical owner of Image 7 may not like the results. Of course, my argument here is using the image structure. In compressed format all this is gone.

4. The authors may want to explore the possibility of combining their approach with newer coding approaches for DNA based storage. This include the use of native DNA (Tabatabaei et al, DOI: 10.1101/672394) and the use of degenerate/composite DNA letters (Choi et al, DOI: 10.1038/s41598-019-43105-w; Anavy et al, DOI: 10.1038/s41587-019-0240-x). Specifically, combining the proposed erasure approach with composite DNA letters may reduce the leaked information to the bare minimum since the encoding is based on the composition in each position rather than using the majority vote to select the base.

We added the following text to the manuscript: "Other encoding schemes, such as using degenerate/composite DNA letters, can also be easily combined with our method." We believe that it's beyond the scope of the current paper to redesign several more panels to show a relatively straightforward combination of our erasure approach with published encoding approach.

>> Agreed

In summary, all this is not necessarily an obstacle to the publication of the manuscript. We do feel, however, that these issues should be addressed and emphasized as part of the manuscript. Doing so will greatly improve the quality of the manuscript, making it a better fit for publication.

Reviewer #1 Additional Comments:

"1. With respect to alternative approaches - how about flushing the DNA solution down the sink? True - some detergent needs to be used in order to guarantee minimal residue but this is still very simple ...

We thank the reviewer for pointing out this alternative method for DNA destruction. This also helped us clarify the unique advantage of our approach, that we can achieve rapid erasure of information in decentralized settings, e.g. during physical transport of the encoded information solution from the sender to a receiver.

>> OK. This should be emphasized in the abstract and the introduction and addressed in the discussion. Including clearly listing alternatives that may work in other situations.

2. With respect to image reconstruction - the following is an example of many papers describe approaches that require very small fraction of remaining pixels:

https://www.researchgate.net/publication/257731990_A_fast_novel_algorithm_for_salt_and_pepper_image_noise_cancellation_using_cardinal_B-splines

The images erased by the authors will be easily recovered by such approaches."

Again, the messages of the image are shown only for visualization purposes. The paper mentioned by the reviewer will be unable to losslessly restore an image from 7% correct pixel blocks. Note that we have added the following text: "In principle, error-correcting codes can be designed so that information is restorable from a very minute fraction of the original information, at the cost of low information content and encoding inefficiency. However, we expect that for actual use when the information solution will be designed to easily erasable via heating, e.g. requiring 80% of original messages to be present for lossless restoration."

>> I accept. But this point should be clarified. It clearly resonates with Reviewer #2 as well ...

In summary:

Some more careful work needs to be done, in my opinion, to clarify and define the application (reassure during physical transfer and not necessarily at either end), the alternatives (from the trivial oven and sink to the more sophisticated chemistry), the framework (compressed messages and NOT images as described). The methods are

nice, innovative, carefully executed and described. This work merits publication but needs to address context more properly.

I apologize to the authors and to the editors for the long delay in my response.

Schedules have somewhat altered ...

Reviewer #2 (Remarks to the Author):

In revising their manuscript, the authors have improved the manuscript, but many of the questions / comments of the referees in the first round of review were not fully answered, and did not result in an improvement of the manuscript. Obviously these input is outside the capacity and scope of the authors. Still, I believe that the manuscript should be published, especially if the referee reports are published together with the manuscript (as routinely done by Nat Commun). However, the two points below are false statements, and should be corrected prior to acceptance.

1. decay of DNA with other methods: The authors write that significant infrastructure is required to destroy DNA with other methods. This is just untrue, and also reflected in the supporting information Figure S6_1: just adding an enzyme, and adjusting the buffer (this can also be done by addition of buffer molecules and does not require buffer replacement) is sufficient to completely destroy the DNA. Also one could think of many other chemical methods for destroying the DNA, involving hydrogen peroxide, heavy metals, strong oxidizing agents (e.g. potassium permanganate), and even household bleach. Chemically DNA is a sensitive molecule, and destroying it is not difficult, even if only using standard chemicals. There are also commercial chemicals developed specifically for the purpose of degrading DNA (e.g. DNAzap), and these are very easy to use, and don't require any infrastructure.

In short: there are very rapid and easy chemical and biological ways to destroy information stored in DNA, they are just not as elegant as the method described in the manuscript. The text in the introduction should reflect this.

We thank the reviewer for pointing out these alternative methods for DNA destruction, and have mentioned these in our revised manuscript. This also helped us clarify the unique advantage of our approach, that we can achieve rapid erasure of information in decentralized settings, e.g. during physical transport of the encoded information solution from the sender to a receiver.

2. it is untrue that fountain codes require 80% of the DNA sequences to be present to successfully decode. The amount of redundancy can be chosen at will during encoding. This can be chosen so that only a few percent of the DNA sequences are required for decoding.

We feel that we need to clarify the distinction between setting up a message where the intention is to be able to recover from minute traces of information (e.g. by encoding with high redundancy) vs. when we have an encoded message in which we would like

to enable rapid permanent erasure (e.g. those that require 80+% of original bits to enable lossless recovery). Paired with our commentary previously about encoding high K-complexity compressed files, we do not see significant risk of recovering the information after heat erasure.

We have added the following text: "In principle, error-correcting codes can be designed so that information is restorable from a very minute fraction of the original information, at the cost of low information content and encoding inefficiency. However, we expect that for actual use when the information solution will be designed to easily erasable via heating, e.g. requiring 80% of original messages to be present for lossless restoration."

Reviewer #2:

Remarks to the Author:

In this round the authors have only slightly improved the quality of the manuscript, and it seems that the authors have spent more effort in debating with the referees than adapting the manuscript.

There are two problems, which remain to be addressed in full prior to publication:

- discuss alternative approaches in destroying DNA, and stating that these might be just as simple to implement (e.g. household bleach, high temperatures, dilution. Some examples in last round of review).

- introduce a better metric on the amount of data lost during decay, and what threshold of remaining information would be theoretically required to reconstruct the data.

Where it seems that the authors have put some thoughts into the second point, the first one remains unresolved (just adding "physical transport" to the text does not resolve this issue.

Notes:

Text in Gray: our responses from last submission.

Text in Green: new reviewers' comments.

Text in Blue: our responses for this resubmission.

Reviewer #1 (Remarks to the Author):

We thank the reviewer for pointing out these alternative methods for DNA destruction, and have mentioned these in our revised manuscript. This also helped us clarify the unique advantage of our approach, that we can achieve rapid erasure of information in decentralized settings, e.g. during physical transport of the encoded information solution from the sender to a receiver.

>> OK. I think that all language should reflect this then. State in the abstract that many possible approaches will destroy the DNA. Including trivial actions such as heating to 250C or flushing in the sink w detergents, as well as chemical approaches such as pointed out by Reviewer #2. The intro should also mention some of these with references. Then the situations where the methods proposed are useful should be pointed out.

We thank the reviewer for pointing out these alternative methods for DNA destruction and have added the following text to the introduction:

"Conventional methods to destroy DNA include irradiating with ultraviolet light, using enzymes such as DNase I [8], applying high temperature of over 200 °C [9, 10], or using bleach [11]. These DNA destruction methods vary in their approach, but are generally difficult to implement in decentralized settings without specialized equipment, and erasure may not always complete within a reasonable timeframe (Supplementary Section S6) [10, 11]. For example, to prevent interception during physical transport of a DNA information solution from one location to another, it may be necessary to have a rapid and complete method for erasing information without specialized equipment for fragile enzymes."

We thank the reviewer for pointing out the potential confusion. We have added the following clarification text: "The different widths and heights of each of the 8 images results in some pixel blocks being resembling the original image (e.g. the right side of Image 6) because there are no false messages in some of the pixel blocks. However, this is an artifact of our selection of images to encode; in actual use of our approach to encode information, it is easy to pad all images with random messages outside the regions of interest so that there are no obvious decoding of some pixel blocks."

>> I accept the response and the text.

Thanks.

We have added text to clarify that fractional remnant information does not represent a security risk: "Erasure of information, in our context, does not require destroying the potential messages by chemical degradation of DNA, but rather relies on the randomization of true/false flags encoded in hybridization state. Given a DNA pool with N addresses and M potential messages at each address, there are M^N possible files; the hybridization states of the $N \cdot M$ DNA oligos inform which one of the M^N messages is the intended message. Once hybridization state information is lost, it is impossible to identify the original message among all the possibilities." See also below regarding our response to low fraction of remaining information.

We disagree with this comment, the space of "reasonable" files should in fact be $(M+1)^N$, because we envision that most users would use a compressed representation of information

(e.g. zipfile) with high K-complexity. For this paper, we only showed bitmap encoding because it is easy for visualize.

We have added the following clarification text: " Here we use the bitmap artwork images to demonstrate our approach in a way that is easy to visualize; however, we do not require any long-range pattern in the underlying data, and we imagine that practical uses of this approach would be on compressed information files with high K- complexity."

We feel that we need to clarify the distinction between setting up a message where the intention is to be able to recover from minute traces of information (e.g. by encoding with high redundancy) vs. when we have an encoded message in which we would like to enable rapid permanent erasure (e.g. those that require 80+% of original bits to enable lossless recovery). Paired with our commentary previously about encoding high K-complexity compressed files, we do not see significant risk of recovering the information after heat erasure.

We have added the following text: "In principle, error-correcting codes can be designed so that information is restorable from a very minute fraction of the original information, at the cost of low information content and encoding inefficiency. However, we expect that for actual use when the information solution will be designed to easily erasable via heating, e.g. requiring 80% of original messages to be present for lossless restoration."

>> OK. See in other comments

Thanks, we have responded to the reviewer's other comments.

We consider the p-values shown in Fig. S5-7 a very strong result, because we have already down-selected from the 8^{76800} to simply 8 potential images. In a real use setting, all 8^{76800} potentially messages would be viewed as equally likely.

>> An insufficient p-value does not mean that you accept the null. It means you can not reject it.

At the suggestion of the reviewer, we have performed additional statistical analysis to test the null hypothesis that the Erased Image of a particular image has Delta fraction more than matched pixels above that of Erased Images deriving from other images. Based on this method, we determine that no more than 2% of the original information remains at a p-value of 0.05, and that no more than 6% of the original information remains at a p-value of 0.001. We have included a snapshot of additional work shown in Supplementary Section S5 below.

FIG. S5-8: Computed p -values for evaluating whether more than Delta fraction of the information remains in the erased images, based on analysis of the matching matrix in Table S5-1.

As an alternative method of analyzing the maximum possible information that remains in the erased images, we computed the p -values for one-way t -tests on the null hypothesis that $M(i,i) \geq \text{Mean}(M(i,j)) + \text{Delta}$, where $M(i,j)$ corresponds to the fraction of matched pixels in row i and column j in Table S5-1. Delta is the hypothesized information that remains, for example the calculation for Delta = 0.05 evaluates the likelihood that $M(1, 1)$ has a true value at least 0.05 higher than the mean of $M(1, j)$, with $j \neq 1$, implying that 5% of the original information has been retained. The computed p values for different values of Delta are plotted for all 8 images in Fig. S5-9. The p values for all 8 images are below $\alpha = 0.05$ when Delta ≥ 0.02 , indicating that no more than 2% of the original information remains in the erased images.

Note that the Delta values implied by fixed p cutoffs represent an upper bound on the mean amount of information retained. As seen in Fig. S5-7, several of the Erased Images actually have smaller fraction of pixels matching to their original image than Erase Images derived from other original images, suggesting that confident identification of the original image is not possible for these particular Erased Images.

As explained previously, it is one thing to recover a message when you know a-priori that there are only 8 possible messages. In practice, there are 8^{76800} possible messages, and we do not realistically think recovering from a minute amount of remaining information is possible.

>> For images this is not true. We can recover from very little information left. There is an issue w the use of 8^{76800} as the number of potential messages. Of course, a compressed image will be much more difficult to recover. But this is not what the authors are demonstrating. The point of showing all the images is a bit mute when its actually not what is meant. Analysis of the compressed case should be added and the images should be less emphasized. An indication of the fact that they are only there for illustration purposes (which the authors state above: " Here we use the bitmap artwork images to demonstrate our approach in a way that is easy to visualize; however, we do not require any long-range pattern in the underlying data, and we imagine that practical uses of this approach would be on compressed information flies with high

K-complexity.") should be added. Furthermore - I do accept the argument re compressed messages but this argument should be made as a main argument. All arguments based on, or illustrated by, the images should become secondary. Otherwise the manuscript risks serious criticism of some claims that can not be applied to the images.

We thank the reviewer for pointing out the potential confusion, and have added the following paragraph to the Discussion:

"The messages of the images are shown only for visualization purposes, and do not represent realistic information encoding schemes that would likely have higher K-complexity than bitmap images. In principle, error-correcting codes can be designed so that information is restorable from a very minute fraction of the original information, at the cost of low information and encoding inefficiency. However, we expect that for actual use the information solution will be designed to easily erasable via heating, e.g. requiring 80% of original messages to be present for lossless restoration."

As described previously, because all of the images encoded have different dimensions, there are certain pixel blocks that do not have corresponding false messages, so the rank would the correct pixel would be 1. This is not an intrinsic limitation of the system, but rather an artifact of the images we selected to encode.

We believe that Fig. S5-7 already sufficiently shows the difficulty of recovering the images. If the reviewer is interested, we would be happy to provide the NGS results in blinded format against the 8 erased images, and invite the reviewer to match each of these to the 8 original images.

>> Thanks for the invitation. The authors may want to propose a student project where the students will be asked to reconstruct Image 7 from the pixels provided. (as a side remark – the mapping btwn the images in F4 to the numbers in F5 is not clear to me). The hypothetic owner of Image 7 may not like the results. Of course, my argument here is using the image structure. In compressed format all this is gone.

We thank the reviewer for the suggestion, but are not currently in a position to implement the suggestion. The PI of this paper, Dr. Zhang, currently teaches two courses at Rice University: an introductory course on biostatistics, and an elective course on biotech and life sciences startups. The proposed project is not a good fit for either course. However, Dr. Zhang will consider putting the t-tests for similarity/differences across the erased images as a homework problem in the statistics course.

We added the following text to the manuscript: "Other encoding schemes, such as using degenerate/composite DNA letters, can also be easily combined with our method." We believe that it's beyond the scope of the current paper to redesign several more panels to show a relatively straightforward combination of our erasure approach with published encoding approach.

>> Agreed

Thanks.

We thank the reviewer for pointing out this alternative method for DNA destruction. This also helped us clarify the unique advantage of our approach, that we can achieve rapid erasure of

information in decentralized settings, e.g. during physical transport of the encoded information solution from the sender to a receiver.

>> OK. This should be emphasized in the abstract and the introduction and addressed in the discussion. Including clearly listing alternatives that may work in other situations.

As described in the response to an above comment, we have added the following text to our introduction:

"Conventional methods to destroy DNA include irradiating with ultraviolet light, using enzymes such as DNase I [8], applying high temperature of over 200 °C [9, 10], or using bleach [11]. These DNA destruction methods vary in their approach, but are generally difficult to implement in decentralized settings without specialized equipment, and erasure may not always complete within a reasonable timeframe (Supplementary Section S6) [10, 11]. For example, to prevent interception during physical transport of a DNA information solution from one location to another, it may be necessary to have a rapid and complete method for erasing information without specialized equipment for fragile enzymes."

Again, the messages of the image are shown only for visualization purposes. The paper mentioned by the reviewer will be unable to losslessly restore an image from 7% correct pixel blocks. Note that we have added the following text: "In principle, error- correcting codes can be designed so that information is restorable from a very minute fraction of the original information, at the cost of low information and encoding inefficiency. However, we expect that for actual use when the information solution will be designed to easily erasable via heating, e.g. requiring 80% of original messages to be present for lossless restoration."

>> I accept. But this point should be clarified. It clearly resonates with Reviewer #2 as well ...

As mentioned previously, we have added the following paragraph to the Discussion, which we believe addresses this concern:

"The messages of the images are shown only for visualization purposes, and do not represent realistic information encoding schemes that would likely have higher K-complexity than bitmap images. In principle, error-correcting codes can be designed so that information is restorable from a very minute fraction of the original information, at the cost of low information and encoding inefficiency. However, we expect that for actual use the information solution will be designed to easily erasable via heating, e.g. requiring 80% of original messages to be present for lossless restoration."

In summary:

Some more careful work needs to be done, in my opinion, to clarify and define the application (reassure during physical transfer and not necessarily at either end), the alternatives (from the trivial oven and sink to the more sophisticated chemistry), the framework (compressed messages and NOT images as described). The methods are nice, innovative, carefully executed and described. This work merits publication but needs to address context more properly. I apologize to the authors and to the editors for the long delay in my response. Schedules have somewhat altered ...

We hope our responses address all remaining concerns of the reviewer.

=====

Reviewer #2 (Remarks to the Author):

In this round the authors have only slightly improved the quality of the manuscript, and it seems that the authors have spent more effort in debating with the referees than adapting the manuscript.

There are two problems, which remain to be addressed in full prior to publication:

- discuss alternative approaches in destroying DNA, and stating that these might be just as simple to implement (e.g. household bleach, high temperatures, dilution. Some examples in last round of review).

We thank the reviewer for pointing out these alternative methods for DNA destruction and have added the following text to the introduction:

“Conventional methods to destroy DNA include irradiating with ultraviolet light, using enzymes such as DNase I [8], applying high temperature of over 200 °C [9, 10], or using bleach [11]. These DNA destruction methods vary in their approach, but are generally difficult to implement in decentralized settings without specialized equipment, and erasure may not always complete within a reasonable timeframe (Supplementary Section S6) [10, 11]. For example, to prevent interception during physical transport of a DNA information solution from one location to another, it may be necessary to have a rapid and complete method for erasing information without specialized equipment for fragile enzymes.”

- introduce a better metric on the amount of data lost during decay, and what threshold of remaining information would be theoretically required to reconstruct the data.

We have added the following paragraph to the Discussion, to reduce potential confusion about image restoration based on a small fraction of remaining information:

"The messages of the images are shown only for visualization purposes, and do not represent realistic information encoding schemes that would likely have higher K-complexity than bitmap images. In principle, error-correcting codes can be designed so that information is restorable from a very minute fraction of the original information, at the cost of low information and encoding inefficiency. However, we expect that for actual use the information solution will be designed to easily erasable via heating, e.g. requiring 80% of original messages to be present for lossless restoration."

At the suggestion of the reviewer, we have performed additional statistical analysis to test the null hypothesis that the Erased Image of a particular image has Delta fraction more than matched pixels above that of Erased Images deriving from other images. Based on this method, we determine that no more than 2% of the original information remains at a p-value of 0.05, and that no more than 6% of the original information remains at a p-value of 0.001. We believe that the "Delta" value, which is a very conservative overestimate of the remaining information fraction, satisfies the reviewer's request for a metric of remaining information.

Where it seems that the authors have put some thoughts into the second point, the first one remains unresolved (just adding "physical transport" to the text does not resolve this issue.

We are open to any additional suggestions for wording that the reviewer may have.